# Subpathway Analysis of Transcriptome Profiles Reveals New Molecular Mechanisms of Acquired Chemotherapy Resistance in Breast Cancer

**DOI:** 10.3390/cancers14194878

**Published:** 2022-10-05

**Authors:** Yang Huo, Shuai Shao, Enze Liu, Jin Li, Zhen Tian, Xue Wu, Shijun Zhang, Daniel Stover, Huanmei Wu, Lijun Cheng, Lang Li

**Affiliations:** 1School of Informatics, Indiana University, Indianapolis, IN 46032, USA; 2Department of Biomedical Informatics, College of Medicine, The Ohio State University, Columbus, OH 43210, USA; 3Department of Medicine, School of Medicine, Indiana University, Indianapolis, IN 46032, USA; 4Division of Medical Oncology, Department of Medicine, The Ohio State University, Columbus, OH 43210, USA; 5Department of Health Service Administration and Policy, College of Public Health, Temple University, Philadelphia, PA 19122, USA

**Keywords:** breast cancer chemotherapy resistance, pathway analysis, transcriptome

## Abstract

**Simple Summary:**

The use of diverse omics platforms and small sample sizes in current studies of cancer chemoresistance limits consensus regarding the molecular mechanisms underlying chemoresistance and the applicability of those study findings. We built transcriptome data for chemotherapy-resistant breast cancer samples for two cohorts and conducted pathway and subpathway analyses that revealed the activation of several molecular pathways associated with chemoresistance in Cushing’s syndrome, human papillomavirus infection, proteoglycans in cancer, fluid shear stress, and focal adhesion that have not been reported in the resistance of breast cancer to chemotherapy. However, analysis of a subset of triple-negative breast cancer samples revealed activation of the identical chemoresistance pathways.

**Abstract:**

Chemoresistance has been a major challenge in the treatment of patients with breast cancer. The diverse omics platforms and small sample sizes reported in the current studies of chemoresistance in breast cancer limit the consensus regarding the underlying molecular mechanisms of chemoresistance and the applicability of these study findings. Therefore, we built two transcriptome datasets for patients with chemotherapy-resistant breast cancers—one comprising paired transcriptome samples from 40 patients before and after chemotherapy and the second including unpaired samples from 690 patients before and 45 patients after chemotherapy. Subsequent conventional pathway analysis and new subpathway analysis using these cohorts uncovered 56 overlapping upregulated genes (false discovery rate [FDR], 0.018) and 36 downregulated genes (FDR, 0.016). Pathway analysis revealed the activation of several pathways in the chemotherapy-resistant tumors, including those of drug metabolism, MAPK, ErbB, calcium, cGMP-PKG, sphingolipid, and PI3K-Akt, as well as those activated by Cushing’s syndrome, human papillomavirus (HPV) infection, and proteoglycans in cancers, and subpathway analysis identified the activation of several more, including fluid shear stress, Wnt, FoxO, ECM-receptor interaction, RAS signaling, Rap1, mTOR focal adhesion, and cellular senescence (FDR < 0.20). Among these pathways, those associated with Cushing’s syndrome, HPV infection, proteoglycans in cancer, fluid shear stress, and focal adhesion have not yet been reported in breast cancer chemoresistance. Pathway and subpathway analysis of a subset of triple-negative breast cancers from the two cohorts revealed activation of the identical chemoresistance pathways.

## 1. Introduction

### 1.1. Genomics Studies of Chemoresistance in Breast Cancer

Recently published U.S. breast cancer statistics [1] indicate that about one in eight women in the U.S. will develop invasive breast cancer, and diagnosis of an estimated 276,480 new cases was expected in 2020. Triple-negative breast cancer (TNBC) accounts for 15 to 20% of cases and a disproportionate 35% of breast cancer deaths [2]. Chemotherapy has been the primary treatment for patients with TNBC, and neoadjuvant chemotherapy (NACT) for early-stage TNBC can make breast-conserving surgery more feasible, providing important prognostic information based on response. In large clinical trials, approximately half of patients with TNBC have residual cancer after NACT [3,4], and approximately 40% of those with residual disease will develop distant metastasis [5]. Targeted treatment options are limited for patients whose TNBC becomes resistant to chemotherapy, despite recent FDA approvals of chemo-immunotherapy for cancer that is PDL1+ [6], PARP1 inhibitors olaparib and talazoparib for patients with BRAC1/2 mutations [7,8,9], and antibody-drug conjugates for late-line therapy [10]. Several ongoing Phase III clinical trials built upon compelling results of Phase II studies target androgen receptors (AR)(NCT02750358), immune checkpoints, such as PD-L1 (NCT02954874, NCT03036488), TROP2 (NCT03498716, NCT03197935, NCT03281954, NCT03125902, NCT03371017), and AKT (NCT03337724). One Phase II study showed a 19% benefit rate with the AR inhibitor, enzalutamide [11], and a second demonstrated a 33.3% response rate using the TROP2 inhibitor, sacituzumab [10]. Another trial investigating the AKT inhibitor, ipatasertib, claimed improved median progression-free survival (PFS) of 6.2 months compared to 4.9 months among patients receiving standard chemotherapies [12]. Nevertheless, potential benefit rates of 19 to about 33% are rather modest, and while we certainly hope for the success of the Phase III clinical trials of these drugs, more translational research is critically needed to identify new targets or drugs for patients with chemotherapy-resistant breast cancers.

Most patients with breast cancer undergo NACT, and response to chemotherapy is assessed at the time of surgical pathology as either pathologic complete response (pCR) or residual disease (RD), and RD is further defined using Miller–Payne or residual cancer burden (RCB) scoring [13]. Patients with complete response do not typically receive additional therapy and are monitored for tumor recurrence, which occurs in only 10 to 15% of patients [14,15]. On the other hand, the near-50% risk of recurrence for patients with residual disease after NACT has generated great interest in evaluating whether the addition of other agents might improve pCR rates and long-term outcomes. Transcriptome profiling at diagnosis (i.e., baseline), right after NACT, or at recurrence reflects dramatically different states of chemotherapy resistance. In a patient with residual disease after chemotherapy, the baseline transcriptome represents the predisposed intrinsic resistance before NACT, whereas the transcriptome right after NACT shows the chemotherapy-induced intrinsic resistance. The transcriptome at recurrence, i.e., acquired resistance, is much more complicated and may be attributable to chemotherapy after a period of dormancy, follow-up radiation, or other cancer therapies.

Table 1 delineates several clinical genomic studies that attempted to investigate the mechanisms of chemotherapy resistance among patients with breast cancer. These studies differed completely with respect to primary clinical endpoints, resistance mechanisms, or genomic platforms. Limited sample sizes also constrained the statistical power of their findings. Consequently, neither gene targets nor biomarkers overlapped among these studies of chemotherapy resistance in TNBC.

### 1.2. Pathway Analysis of Genomics Profiles

Pathway analysis is an important method for exploring the functions of signaling pathways using transcriptomic data, and gene set enrichment analysis (GSEA) is the most classical approach to pathway analysis [25]. GSEA tests whether differentially expressed genes are overexpressed in a particular pathway as compared with expression in the rest of the genome [25]. Several computational methods integrate regulation mechanisms into the analysis of pathway activity, taking into consideration the topological structures of the signaling pathways. The review of Ma et al. reports many salient examples of methods to analyze pathway enrichment, including SPIA, CePa, NetGSA, TopologyGSA, DEGraph, CAMERA, PRS, PathNet, and others [26]. Recent developments in pathway analysis have delved further in order to consider subpathways. Amadoz’s team reviewed subpathway analysis methods that integrate topology data, including Hipathia, MinePath, PathiVar, Pathome, Pathiways, DEAP, TopologyGSA, subSPIA, and others [27]. However, despite these advancements, there remains a significant need to improve subpathway analysis. In this paper, our pathway and subpathway analyses focus on signaling transduction. We are interested in whether a molecular signal can be transmitted from upstream to downstream in a pathway or its subpathways, and we use the Kyoto Encyclopedia of Genes and Genomes (KEGG) pathways as examples. None of the existing subpathway analyses clearly or adequately characterize the subpathways from a pathway because the analyses were designed to detect the altered signaling transduction among different conditions rather than the strength of the signaling transduction.

We focus here on acquired chemotherapy resistance in patients with breast cancer, comparing transcriptome profiles before and after chemotherapies. To address the challenge of small sample sizes in previous genomic studies of breast cancer, we have integrated data from various sources, and we hope our pathway and subpathway analyses will overcome the lack of overlapping genes among those earlier studies to offer more robust insight into the mechanisms underlying breast cancer resistance to chemotherapy.

## 2. Methods

### 2.1. Data Collection and Integration

We queried datasets of the Gene Expression Omnibus (GEO) [28] using “breast cancer” and “chemo” as keywords, which generated 7991 breast cancer-related datasets, and downloaded their data description files, each of which included topic, abstract, platform, and sample size. We then wrote a Python program that excluded data from both animal studies and in vitro experiments and then manually reviewed and curated data regarding drug resistance status (Miller–Payne score of one to four signifying drug sensitivity and five reflecting drug resistance), ER/PR/HER2 status, age, and race. We also collected microarray gene expression platforms, including GPL96 and GPL570. In the end, we collected 37 datasets for 815 breast cancer samples from the GEO.

We also queried the ArrayExpress database using the keyword “breast cancer,” which yielded 3975 experiments, excluded data from both animal studies and non-Affymetrix platforms, and then excluded all duplicated IDs in GEO, ultimately collecting 87 breast cancer samples from ArrayExpress.

After collecting all relevant data from GEO and ArrayExpress, we created three cohorts of patient data. The first included 40 paired tumor samples (before and after chemotherapy), each pair obtained from a single patient with breast cancer. The second included unpaired tumor samples (before and after chemotherapy) from different patients matched by age, race, and ER/PR/HER status (Table 2). The third included all the TNBC samples from the previous two cohorts. We chose the Affymetrix U133A microarray platform for data analysis.

### 2.2. Differential Gene Expression Analysis and Pathway Analysis

Analysis of differential gene expression. A gene will be called “present” if at least one probe is present [29]. We analyzed gene expression before and after chemotherapy by comparing the odds of gene presence over gene absence. We used McNemar’s test when the dataset comprised paired transcriptome data from the same patient [30] and the chi-square test for comparison in a non-paired transcriptome dataset [31]. We also calculated the fold-change of gene expression, with a fold-change higher than one indicating the upregulation of gene expression after chemotherapy.

Downloading KEGG pathways: The KEGG pathway database provides an R package (KEGGx) for processing .hxml files, which we used to download 294 pathways. We then removed compound-only and non-human pathways, finally selecting 65 pathways for this project (Appendix A).

We performed pathway enrichment analysis on overlapping genes between the 14,529 genes in the Affymetrix microarray platform and the 8988 genes in 65 pathways and performed hypergeometric distribution-based enrichment analysis to test whether expression of a particular gene was greater in a given pathway than its expression in all other pathways. The particular genes undergoing analysis were chosen from lists of either differentially expressed, upregulated, or downregulated genes.

### 2.3. Analysis of Subpathway Data

Subpathway definitions: A single-chain signaling pathway starts from one entry node (i.e., a gene node without parent) and ends in an end node (i.e., a gene node without a child). The creation of such a single-chain signaling subpathway depends on the topology of the pathway. Figure 1 illustrates three different schemes for defining single-chain signaling subpathways—canonical, binding, or looping.

The subpathways of a canonical pathway (Figure 1A) are composed of all the possible paths from a node to an end node. In a pathway with a binding event (Figure 1B), i.e., B and C, three different subpathways are created—(A, B, D), (A, C, D), and (A, B+C, D). This setup reflects our assumption that this pathway is active if at least one of B or C is present when B and C have a binding event. Figure 1C characterizes how a subpathway is created when there is a looping event, B-C-D-E-B. We lump all the genes in the loop together into one node if all the genes in the node are active. Otherwise, the loop will not be active.

Activation and inhibition: Table 3 shows the 16 types of relationships delineated in KEGG grouped into two relationship types—induction (defined as 1) and inhibition (defined as −1) [32,33].

Calculation of subpathway impact score. The perturbation factor (PF) of the gene g is calculated in Equation (1):(1)$$PF\left(g\right)=\Delta E\left(g\right)\cdot \chi^{2}+\beta_{ug}\cdot PF\left(u\right)$$

In Equation (1), ΔE(g) represents a log-fold change of gene expression after chemotherapy over its expression before chemotherapy, and χ is the chi-square statistic calculated by either McNemar’s test (paired dataset) or chi-square statistics (unpaired sample dataset). The second term on the right-hand side of Equation (1) is the PF of the gene u directly upstream of the target gene g. It is weighted by the factor $\beta_{ug}$, which indicates the type of interaction: $\beta_{ug}=1$ for induction and −1 for inhibition, as shown in Table 2.

The impact factor of this subpathway is calculated in Equation (2):(2)$$IF\left(subpath\right)=log\left(\frac{1}{P_{path}}\right)+\frac{\sum_{g\in subpath}\left|PF\left(g\right)\right|}{\left|\bar{\Delta E}\right|\times N_{de}\left(subpath\right)}$$

In Equation (2), $P_{path}$ represents the probability of hypergeometric distribution of the number of significant genes over the number of genes in the pathway. PF is the result of Equation (1). $N_{de}$ is the number of differentially expressed (DE) genes in a subpathway, and a larger IF score indicates the greater importance of the subpathway.

Empirical *p*-value for subpathway impact scores: The gene expression samples are randomly permutated between before and after chemotherapy. In the paired set, the before and after samples are permutated in the same patient, the permutated samples undergo subpathway analysis, and subpathway impact factor scores are calculated for each subpathway. This permutation analysis is performed 10 times. An empirical distribution of subpathway impact factor scores is then formed for a set of subpathways with the same length, and the *p*-value of our designated subpathway analysis is calculated as the tail probability, i.e., percentile, from the empirical distribution with the same length.

## 3. Results

### 3.1. Genes Expressed Differentially between Pre- and Post-Chemotherapy among Patients with Breast Cancer

In the paired-sample breast cancer cohort, post-chemotherapy samples demonstrated significant upregulation of 152 genes (*p* < 0.05) and significant downregulation of 112 genes compared with expression in their paired pre-chemotherapy samples (Figure 2). In the unpaired sample cohort, 1616 genes were significantly upregulated, and 1108 genes were significantly downregulated. The paired and unpaired cohorts together demonstrated the upregulation of 56 overlapped genes (FDR, 0.018) and the downregulation of 36 (FDR, 0.016) (Appendix A). Further analysis of pathway enrichment among the upregulated genes in the paired and unpaired samples demonstrated enrichment in both paired and unpaired samples among their overlapped signaling pathways, including drug metabolism, MAPK, ErbB, calcium, cGMP-PKG, sphingolipid, PI3K-Akt, Cushing’s syndrome, HPV infection, and proteoglycans in cancers (*p* < 0.05) (Table 4).

### 3.2. Pathways and Their Statistically Significantly Activated Subpathways from Pre- to Post-Chemotherapy in Both Paired and Unpaired Breast Cancer Sample Cohorts

Subpathway impact analysis was performed separately for both paired and unpaired chemotherapy-resistant breast cancer samples; findings of the two analyses were combined; the overlapping upregulated subpathways were selected, and their false discovery rates were calculated. Table 5 details 12 pathways with significantly overlapped and upregulated subpathways (FDR < 0.20). In each pathway and its overlapped subpathways, we further reported significantly upregulated genes.

One striking finding is that most of the activated subpathways in every pathway are well-connected and often densely concentrated.

### 3.3. Pathway and Sub-Pathway Analyses of Chemoresistance in Triple-Negative Breast Cancer

In the TNBC samples (Table 6), 2358 genes were significantly upregulated (*p* < 0.05), and 1527 genes were significantly downregulated. Taking the up- and downregulated samples together, we performed the hypergeometric test for pathway enrichment analysis and its follow-up subpathway analysis. Table 5 reports 12 pathways with enriched activated subpathways and significant impact score (*p* < 0.01, FDR < 0.47), which coincide with the top 12 pathways (Table 4) enriched in the paired and unpaired chemo-resistant breast cancer cohorts. The FDR is noticeably larger than that for the chemotherapy-resistant breast cancer because it involves only one cohort.

## 4. Discussion and Conclusions

Using two cohorts of chemotherapy-resistant breast cancer tumor samples, one paired cohort and one unpaired cohort, we identified the upregulation of 56 genes (FDR, 0.018) and downregulation of 36 (FDR, 0.016) in tumors that have become resistant to chemotherapy. The upregulated genes showed enrichment in ten pathways (Table 4), many of which have been reported in studies of breast cancer chemoresistance. The PI3K/AKT pathway is probably the most frequently studied in this regard. Li et al. have observed that its inhibition can overcome breast cancer resistance to doxorubicin [34], and its activation via the upregulation of the ErbB signaling pathway has been reported to lead to multi-drug resistance in breast cancer [35,36]. By inhibiting the drug efflux transporter ABCC1, LINC00518 has also been identified to overcome chemoresistance in breast cancer [37], driven by sphingosine kinase-1 (SPHK1), which regulates the sphingolipid signaling pathway [38]. The MAPK pathway is also widely associated with this chemoresistance. Christowitz’s research group found that the upregulation of the MAPK/ERK pathway underlies doxotubicin-induced drug resistance [39], and Hasna et al. reported that the overexpression of Orai3 calcium channels can downregulate expression of the p53 tumor suppressor protein, leading to chemoresistance in breast cancer [40]. Furthermore, upregulation of MDR1 expression by both the Wnt/β-catenin pathway and the cAMP signaling pathway affect chemoresistance in breast cancer [41,42,43]. Nevertheless, our pathway analysis uncovered three new genetic pathways not previously studied in breast cancer chemoresistance, including Cushing’s syndrome, human papillomavirus infection, and proteoglycans in cancer, that deserve more investigation.

Using our newly developed subpathway analysis, we have identified many pathways whose subpathways are activated in chemotherapy-resistant breast cancer, including those of fluid shear stress and atherosclerosis, as well as Wnt, FoxO, ECM-receptor interaction, RAS signaling, Rap1, mTOR, and cellular senescence. Some of these have been reported in the literature, with PI3K/AKT and mTOR being the most-reported pathways associated with breast cancer chemoresistance. Many PI3K, AKT, and mTOR inhibitors have been designed, and clinical studies are in progress [44,45,46]. An essential role of the ECM-receptor interaction pathway has been identified in doxorubicin treatment of breast cancer in an in vitro model [47], and fluid shear stress and atherosclerosis pathways have been shown to induce resistance to chemotherapy drugs and proliferative tendencies in a cell-line model of breast cancer [48]. Moreover, oxidatively stressed multi-nucleated cells (MNC) have been shown to induce chemoresistance in TNBC in in vitro and in vivo models by activating the RAS/MAPK pathway [49], and cellular senescence has been identified as a potential mechanism of chemoresistance in TNBC [50]. The upregulation of the Ras/PI3K/PTEN/AKT/mTOR pathway and the Ras/Raf/MEK/ERK pathway together can lead to chemoresistance by diminishing cell senescence [51]. FoxO1 has been demonstrated to increase the expression of MDR1, leading to breast cancer chemoresistance [52], and based on our subpathway model, the *FoxM1* gene is the most significant in activating the cAMP signaling pathway, proving to be a promising candidate target for treating this chemoresistance [53]. Rap1, fluid shear stress and atherosclerosis, and focal adhesion can be considered new pathways of chemoresistance, not having been reported in the drug resistance literature related to breast cancer.

Another interesting result of subpathway analysis is that all subpathways activated in each pathway are well-connected and condensed among hub genes, i.e., updated genes that appear repeatedly in many subpathways (Figure 3, Table 5). Some of these hub genes, such as *FoxM1* [53] and *FoxO1* [52], have already been chosen as drug targets; the other 28 represent potential targets for overcoming chemotherapy resistance in breast cancer.

Chemoresistance in triple-negative breast cancer reveals almost identical pathways and activated subpathways (Table 6) when compared to breast cancer chemoresistance pathway analysis. However, our analysis of chemoresistance in TNBC includes only one cohort of samples that integrates paired and unpaired TNBC samples from two breast cancer cohorts because the number of TNBC patient samples in each data cohort was limited. This is one area that deserves further validation.

In this paper, we focused on Affymetrix microarray data rather than data from a next-generation sequencing (NGS) platform because the number of NGS post-chemotherapy breast cancer samples is limited. In Table 1, we enumerated clinical genomic studies of breast cancer resistance. In our referenced articles, we identified 150 NGS samples with targeted RNA−76 primary breast cancer profiles and 74 post-chemotherapy breast cancer profiles. None of them contain paired pre- and post-chemotherapy samples from the same patient, and these 150 NGS samples have different RNA sequencing platforms—112 have targeted RNA sequencing and the remaining 38 have whole transcriptome RNA sequencing. We also queried both GEO and Array-Express and found only 12 pairs of tumor samples with RNA sequencing data. Finally, The Cancer Genome Atlas (TCGA) includes 1207 primary breast cancer RNA-sequencing data before chemotherapy. Thus, whole transcriptome RNA sequencing data is very limited for breast cancer samples post-chemotherapy.

In conclusion, for the first time, using subpathway analysis, we have identified a number of activated subpathways in Cushing’s syndrome, HPV infection, proteoglycans in cancer, fluid shear stress, and focal adhesion pathways in breast cancer chemoresistance.

## Figures and Tables

**Figure 1 cancers-14-04878-f001:**
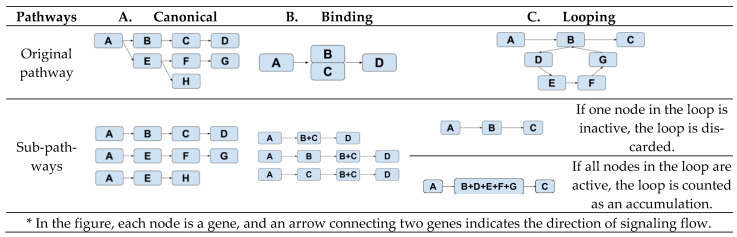
Derivations of subpathways from three topologically different pathways. *

**Figure 2 cancers-14-04878-f002:**
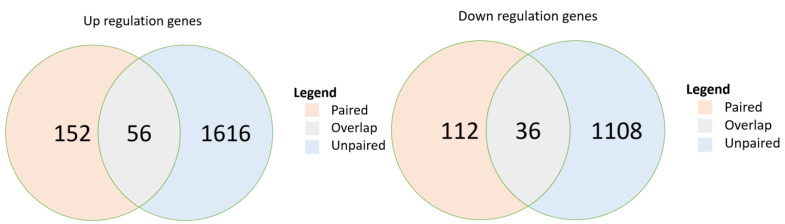
Overlap of differentially expressed genes in two breast cancer cohorts: paired sample cohort and unpaired sample cohort. *p*-values are calculated from hypergeometric distribution; pathways were selected if *p* < 0.05. DE, differentially expressed.

**Figure 3 cancers-14-04878-f003:**
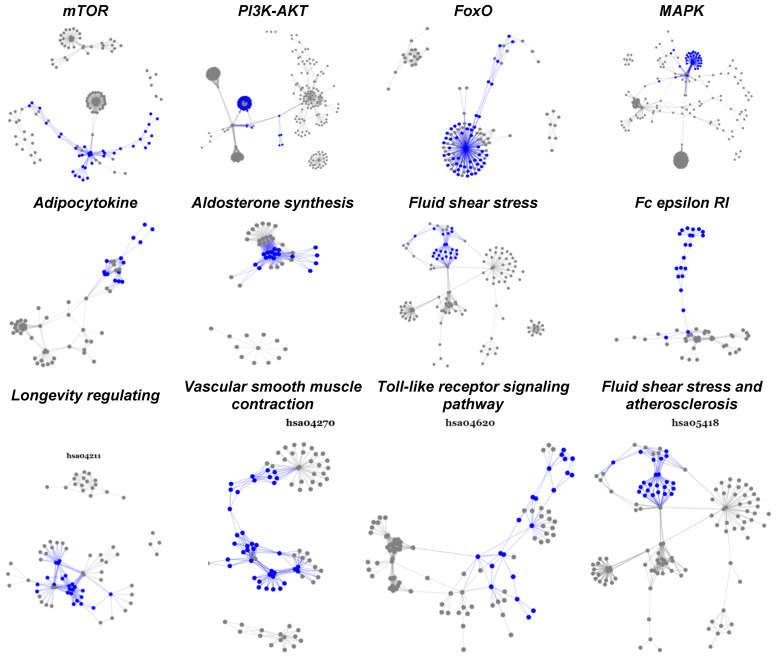
Statistically significant and biologically active subpathways in TNBC chemoresistance. This figure displays twelve significantly activated pathways overlapped between paired and unpaired cohorts. The highlighted parts (blue) are subpathways upregulated as the breast tumor developed from pre- to post-chemotherapy.

**Table 1 cancers-14-04878-t001:** Clinical genomic studies of chemotherapy resistance in breast cancer.

Publications 2010–2018	Clinical Endpoint and Sample Size	Genomic Platform	Primary Genes and Pathways Discovered
Balko et al. 2012 [16] Balko et al. 2014 [17] Balko et al. 2016 [18]	Relapse-free survival (RFS) n = 74	Targeted RNA and DNA sequencing	DUSP4 low expression, MYC high expression, and JAK2 amplification were associated with RFS.
Lips et al. 2015 [19]	Pathologic complete response (pCR) and RFS n = 56	Targeted DNA sequencing	No statistically significant genes
Kim et al. 2018 [20]	pCR was not defined after NACT n = 20	Bulk DNA sequencing, single nucleus RNA- seq/DNA-seq	Chemoresistance gene signatures are enriched in EMT, CDH1, AKT1, hypoxia, angiogenesis, and extracellular matrix degradation signaling pathways.
Laura et al. 2013 [21]	pCR n = 106	Affymetrix	Significant genes enriched in Wnt, HIF1, p53, and Rho GTPases signaling pathways were associated with poor response to chemotherapy drugs.
Korde et al. 2010 [22]	pCR n = 21	Affymetrix	MAP-2, MACF1,VEGF-B, and EGFR showed high expression in patients without pCR after chemotherapy.
Silver et al. 2010 [23]	pCR n = 28	Affymetrix	BRCA1 promoter methylation and E2F3 activation contribute to good cisplatin response.
Stover et al. 2015 [24]	pCR n = 446	Affymetrix, Agilent	Low proliferation and immune-predicted resistance, with stem-like phenotype and Ras-Erk were associated with chemotherapy resistance.

**Table 2 cancers-14-04878-t002:** Matching scheme for the second cohort of breast cancer samples pre- and post-chemotherapy. *

Group Demographics Clinical Status	Pre-Chemo	Percentage (%)	Post-Chemo	Percentage (%)
Age < 55, Race = white ER+ PR+ HER2+	15	2.18	1	2.22
Age < 55, Race = white ER+PR-/ER-PR+/ER-PR-HER2-	89	12.90	5	11.11
Age < 55, Race = white ER+PR-/ER-PR+/ER-PR-HER2+	21	3.04	2	4.44
Age < 55, Race = white ER-PR-HER2-	46	6.67	7	15.56
Age < 55, Race = non-white ER+PR+HER2+	22	3.19	1	2.22
Age < 55, Race = non-white ER+PR-/ER-PR+/ER-PR-HER2-	52	7.53	6	13.33
Age < 55, Race = non-white ER+PR-/ER-PR+/ER-PR-HER2+	35	5.07	4	8.89
Age < 55, Race = non-white ER-PR-HER2-	48	6.96	4	8.89
Age > 55, Race = white ER+PR+HER2+	31	4.49	0	0
Age > 55, Race = white ER+PR-/ER-PR+/ER-PR-HER2-	94	13.62	6	13.33
Age > 55, Race = white ER+PR-/ER-PR+/ER-PR-HER2+	30	4.35	3	6.66
Age > 55, Race = white ER-PR-HER2-	55	7.97	6	13.33
Age > 55, Race = non-white ER+PR+HER2+	51	7.39	0	0
Age > 55, Race = non-white ER+PR-/ER-PR+/ER-PR-HER2-	30	4.34	0	0
Age > 55, Race = non-white ER+PR-/ER-PR+/ER-PR-HER2+	25	3.62	0	0
Age > 55, Race = non-white ER-PR-HER2-	46	6.67	0	0

* The matching scheme delineated in the table allowed us to detect genes expressed differentially between the pre- and post-chemotherapy samples—genes that would not be confounded with the indicated demographic and clinical factors.

**Table 3 cancers-14-04878-t003:** Between-node relationships in pathways.

Name	Relationship	Value
Activation	-->	1
Inhibition	--|	−1
Expression	-->	1
Repression	--|	1
Indirect effect	..>	1
State change	…	1
Binding/association	---	1
Dissociation	-+-	−1
Missing interaction	-/-	1
Phosphorylation	+p	1
Dephosphorylation	−p	−1
Glycosylation	+g	1
Ubiquitination	+u	1
Methylation	+m	1

**Table 4 cancers-14-04878-t004:** Pathway analysis of upregulated genes in paired and unpaired sample cohorts. *

Pathway Name	Number of Genes in Pathway	DE Genes in Paired Samples	DE Genes in Unpaired Samples	*p*-Value (Paired Samples)	*p*-Value (Unpaired Samples)
Drug metabolism	133	13	41	9.74 × 10^−07^	0.00041
MAPK signaling pathway	278	26	37	9.99 × 10^−12^	0.00228
ErbB signaling pathway	131	14	15	1.25 × 10^−07^	0.00615
Calcium signaling pathway	170	20	43	4.44 × 10^−11^	0.01012
cGMP-PKG signaling pathway	188	26	13	9.78 × 10^−16^	1.075 × 10^−06^
Sphingolipid signaling pathway	137	13	40	1.35 × 10^−06^	0.00134
PI3K-Akt signaling pathway	416	16	32	0.00287	1.52 × 10^−11^
Cushing’s syndrome	194	30	17	2.14 × 10^−19^	2.70 × 10^−05^
Human papillomavirus infection	387	22	18	2.43 × 10^−06^	1.81 × 10^−17^
Proteoglycans in cancer	332	16	44	0.000330	0.000977

* This table presents enriched pathways overlapped in both paired and unpaired sample cohorts.

**Table 5 cancers-14-04878-t005:** Significantly upregulated subpathways and their associated genes in both paired and unpaired drug-resistant breast cancer samples.

Pathway Name	Number of Subpathways (*p* < 0.05, Paired)	Number of Subpathways (*p* < 0.05, Un-Paired)	Overlapped Upregulated Subpathways (Same Direction)	False Discovery Rate (Overlap)	Significant Genes (Up-Regulated) *
Fluid shear stress and atherosclerosis	1085	1694	277	0.10	MAP3K5
MAPK signaling pathway	1947	1541	358	0.11	MAP2K2, MAP3K1, MAP3K5, MAP4K1
PI3K-Akt signaling pathway	1470	1325	254	0.13	NR4A1, NRAS, PIK3R3, OSMR
Wnt signaling pathway	1775	2635	315	0.14	MAP2K1, MAPK1
FoxO signaling pathway	1432	2175	246	0.15	FOXO6, FBXO25 FOXO1
ECM-receptor interaction	1033	1876	167	0.15	MYL9, IRS2
Ras signaling pathway	1019	1460	164	0.16	RAC3
Rap1 signaling pathway	861	1505	132	0.16	CTNNB1, MAGI3, RAPGEF6, RAP1B, ARAP3
mTOR signaling pathway	831	1253	126	0.16	MAPK3, GSK3B
Calcium signaling pathway	820	1238	124	0.16	PLCG1, PLCG2, PRKCG
Cellular senescence	684	1341	97	0.18	TP53, CDKN1A
cAMP signaling pathway	1937	2598	247	0.20	E2F1, FOXM1

* The last column includes genes with the top perturbation factor score.

**Table 6 cancers-14-04878-t006:** Significantly upregulated subpathways in drug-resistant samples of triple-negative breast cancer.

Pathway Name	Number of Subpathways	Number of Subpathways (*p* < 0.01)	False Discovery Rate
*Calcium signaling pathway*	12,056	1598	0.075
*Fluid shear stress and atherosclerosis*	12,337	1347	0.091
*cAMP signaling pathway*	19,409	1602	0.121
*Cellular senescence*	18,190	1040	0.174
*mTOR signaling pathway*	19,518	1097	0.177
*MAPK signaling pathway*	30,652	1553	0.197
*ECM-receptor interaction*	15,162	658	0.230
*PI3K-Akt signaling pathway*	34,998	1436	0.243
*Focal adhesion*	22,580	853	0.264
*Wnt signaling pathway*	28,687	835	0.346
*Ras signaling pathway*	40,147	1092	0.367
*Rap1 signaling pathway*	20,606	439	0.469

## Data Availability

Data involved in this research from database KEGG Pathway Database. (https://www.genome.jp/kegg/pathway.html, accessed on 14 August 2021). Detailed selection shown in Appendix A.

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
