# Peer review of "Subpathway Analysis of Transcriptome Profiles Reveals New Molecular Mechanisms of Acquired Chemotherapy Resistance in Breast Cancer"

_cancers, 2022, doi:10.3390/cancers14194878_

Round 1
Reviewer 1 Report
In this submitted research work the authors have tried to explore the prognostic value of genes associated with breast cancer chemo resistance, including Cushing syndrome, human papillomavirus infection, and proteoglycans in cancer. The authors have made excellent use of pathway analysis based of signalling-pathway impact analysis method to elucidate 3 signalling sub-pathways when breast cancer becomes chemo resistant.
However, a major suggestion for the authors would be to try and accumulate/expand more "paired breast cancer chemo resistance cohort" data, to validate the activated sub-pathways elucidated by the authors in this work. It might be a difficult task to gather such data on short distance, but it would be nice to get the identified pathways validated especially the FoxM1, sub-pathway model.
Author Response
Dear reviewers
Thanks very much for taking the time to review this manuscript. We appreciate all your great comments and suggestions!
Reviewer 2 Report
The present study from Huo et al seeks to better understand resistance mechanisms of breast cancer upon chemotherapy as earlier studies were usually limited by small sample size and variable parameters used. Through analyzing integrated publicly available transcriptome datasets on pre and post-chemo breast cancer patients, Huo et al., identified both previously reported and novel genes and pathways are differentially modulated upon chemotherapy. The author further applied a novel sub-pathway impact score to reveal activated sub-pathways upon chemotherapy. Even though this work could provide some valuable insights for the breast cancer research community, there are multiple concerns that need to be carefully addressed before considering the manuscript for publication.
Major comments:
1. Although the author explicitly wrote that there was three timepoints where cancer transcriptome could be assessed (baseline, right after chemo, and at recurrence) and earlier studies usually have different clinical endpoints, it is still confusing to me what pre and post-chemotherapy breast cancer dataset the author used for comparison in this study. Is it baseline vs recurrence or baseline vs right after chemo? Please explain with more details in the method section. It would make more sense if it were baseline vs right after chemo because transcriptome at recurrence can be modulated by other factors, as the author suggested in the manuscript.
2. In your data collection, did the authors only use microarray data in this study? If so, please state it clearly in the first paragraph because it is a very important criterion for the readers to know. It is apparent that more data have been collected in NGS platform due to obvious reasons, I am wondering why the author focuses exclusively on microarray data but leaves NGS data behind? I’d be more convincing if NGS data showed up similar results.
3. It looks like most analysis included in the current study was done in the paired and unpaired but matched tumor samples. However, the authors did include a third group of TNBC samples but without explaining the rationale. But apparently, TNBC is the major group that receives chemotherapy and subsequently becomes refractory. Please add an appropriate amount of descriptions in the manuscript (i.e, how many TNBC patients are included, why you have a third group derived from your first and second group, etc)
4. Table 2 is very confusing, and no cohort information is found. Please reorganize.
5. In Table 3 (between node relationships in pathways), for methylation, I would think methylation could be either induction or inhibition, please explain why the authors only consider induction?
6. In equation 2, from the biological perspective, would it make more sense that larger the divisor (|ΔÌ…Ì…?Ì…|×???(??????â„Ž)) of the second part of equation 2 (stronger expression changes on more genes), the more important this sub-pathway is? However, if that's the case, the bigger the IF(subpath) doesn't necessarily mean more importance.
7. In the results section, it appeared that the cutoff for the differentially expressed gene is a nominal p-value <0.05, why didn’t the author use the corrected p-value? The corrected p-value seems to be a more valid cutoff for differentially expressed gene analysis. Please provide reasoning here.
8. For the 56 overlapped genes are up-regulated and 36 genes are downregulated when combining the paired and unpaired cohorts, the author calculated each group an FDR, why and how did the author calculate this FDR? Similar issues also found in section 3.3 (P <0.01, and FDR < 0.47). The FDR <0.47 seems to be too loose for making a point.
Minor comments:
1. Please explain RFS when it first appeared
2. All supplement tables are missing
3. No figure legend was provided
4. Table 6 should be referred in section 3.3
5. Figure 1 and Table 1 mingled together
Round 2
Reviewer 2 Report
Please see the attached document.
